# Computerised cognitive training in Parkinson's disease: a protocol for a systematic review and updated meta-analysis

Hanna Malmberg Gavelin [ID],[1,2] Magdalena Domellöf [ID],[2] Isabella Leung,[3,4] Anna Stigsdotter Neely [ID],[5] Carsten Finke [ID],[6,7] Amit Lampit [ID] [1,6,7]

For numbered affiliations see end of article.

**Correspondence to**
Dr Amit Lampit;
amit.lampit@unimelb.edu.au

## ABSTRACT

**Introduction** Cognitive impairment is recognised as an important non-motor symptom in Parkinson's disease (PD) and there is a need for evidence-based non-pharmacological interventions that may prevent or slow cognitive decline in this patient group. One such intervention is computerised cognitive training (CCT), which has shown efficacious for cognition across older adult populations. This systematic review aims to investigate the efficacy of CCT across cognitive, psychosocial and functional domains for people with PD and examine study and intervention design factors that could moderate CCT effects on cognition.

**Methods and analysis** Randomised controlled trials investigating the effects of CCT in patients with PD without dementia, on cognitive, psychosocial or functional outcomes, will be included. The primary outcome is overall cognitive function. Secondary outcomes are domain-specific cognitive function, psychosocial functioning and functional abilities. We systematically searched MEDLINE, Embase and PsycINFO through 14 May 2020 to identify relevant literature. Risk of bias will be assessed using the revised Cochrane Risk of Bias tool. Effect sizes will be calculated as standardised mean difference of baseline to postintervention change (Hedges' *g*) with 95% CI for each eligible outcome measure. Pooling of outcomes across studies will be conducted using random-effects models, accounting for dependency structure of effect sizes within studies. Heterogeneity will be assessed using $\tau^2$ and $I^2$ statistic. Potential moderators, based on key study and intervention design factors, will be investigated using mixed-effects meta-regression models.

**Ethics and dissemination** No ethical approval is required. The findings will be disseminated in a peer-reviewed scientific journal.

**PROSPERO registration number** CRD42020185386.

## Strengths and limitations of this study

► Specification of the eligibility criteria for randomised controlled trials of narrowly defined computerised cognitive training delineates the effects of this intervention from other cognitive intervention approaches in Parkinson's disease.

► Inclusion of a variety of cognitive, psychosocial and functional outcome measures will improve statistical power to inform efficacy across and within domains.

► Accounting for dependency of effect sizes within studies will reduce overestimation of within-study variance and thus underestimation of between-study heterogeneity, especially since individual studies are expected to be underpowered.

► Multiple methods for investigating heterogeneity can inform intervention and study design, but contingent on the number and size of available studies.

► Analyses are limited to group—rather than individual participant data.

risk.[1] Considering the negative influence of cognitive impairment on quality of life for patients as well as caregivers and the current lack of effective pharmacological treatments,[1] developing interventions that could maintain cognitive function and delay cognitive and functional decline is a critical area for prevention and treatment research in the field.[2]

Cognitive training is a non-pharmacological intervention that has shown efficacious for cognition in older adults across the spectrum from cognitively healthy to dementia.[3] Specifically, computerised cognitive training (CCT) has received widespread attention in recent years as a safe and scalable intervention that can incorporate important intervention design features such as adaptivity of training difficulty and continuous motivational feedback on training performance.[4] In addition to efficacy for cognition, individual trials have reported potential benefits on other symptoms such as mood and freezing of gait.[4]

## INTRODUCTION

Cognitive decline is one of the most common non-motor symptoms in Parkinson's disease (PD).[1] Approximately 20% of people with PD already have mild cognitive impairment (MCI) at diagnosis, with over 40% conversion to dementia 10 years after PD diagnosis, substantially exceeding age-standardised

To date, several systematic reviews and meta-analyses have investigated the efficacy of cognitive training on cognitive function in PD.[5–8] Leung and colleagues[5] identified seven randomised controlled trials (RCTs) and concluded that cognitive training showed modest efficacy for overall cognition in PD, with larger effect sizes observed within individual cognitive domains.[5] Similar results were reported by Lawrence and colleagues,[6] who combined 11 randomised and non-randomised cognitive training trials. More recently, a Cochrane review identified seven RCTs investigating the effects of cognitive training in patients with PD with MCI or dementia, reporting imprecise and uncertain evidence for efficacy on global cognition.[7] Approaches for estimating effect sizes across studies varied across reviews, and none conducted investigations of heterogeneity. Finally, a recent systematic review focusing specifically on CCT reported evidence for cognitive benefits based on seven RCTs; however, no meta-analysis was performed and potential effect modifiers were not explored.[8]

Taken together, previous reviews have shown mixed and inconclusive results and the efficacy of cognitive training in general, and CCT in particular, in people with PD remains uncertain. Furthermore, given the limited number of studies in previous reviews as well as clinical and methodological heterogeneity, the effects of CCT across different cognitive, psychosocial and functional domains as well as design factors that may be associated with such effects are still unclear.[4]

## Objectives

The aim of this review is to evaluate the efficacy of CCT on cognitive, psychosocial and functional outcomes in persons with PD. Specifically, we aim to:

1. Investigate the efficacy of CCT on cognitive, psychosocial and daily function in PD, in comparison to active or passive control.
2. Examine study and intervention design factors that could moderate CCT effects on cognitive function across studies.
3. Evaluate the strength and quality of the evidence for CCT in PD.
4. Suggest recommendations for future research and practice in the field.

## METHODS AND ANALYSIS

This protocol adheres to the Preferred Reporting Items for Systematic Review and Meta-analysis Protocols (PRISMA-P) guidelines[9] and the protocol was prospectively registered with PROSPERO. The PRISMA-P checklist can be found in the online supplemental file 1. This review updates and further specifies our previous systematic review on cognitive training in PD.[5]

## Eligibility criteria

Consistent with our previous systematic reviews of CCT,[10–12] we will include studies that meet the following criteria:

### Types of studies

RCTs studying the effects of CCT on one or more cognitive, psychosocial or functional outcome in patients with PD. Eligible studies will provide neuropsychological testing at baseline and post-CCT intervention. Randomised crossover trials will be included, but only the first treatment phase will be considered and used for analysis. Non-randomised trials will be excluded. Unpublished RCTs or those published as conference abstracts, theses or monographs will be eligible if data needed for analysis and appraisal can be obtained from the authors.

### Types of participants

Patients with PD (any age and aetiology), either cognitively healthy, with subjective cognitive decline or MCI. Studies including only or mainly people with dementia will be excluded. Studies reporting the results from a mixed population (eg, MCI and dementia) will be eligible if the results for the eligible population are reported or can be obtained separately or if the eligible population (eg, PD with MCI) constitutes ≥50% of the sample.

### Types of interventions

Minimum of 4 hours of practice on standardised computerised tasks or video games with clear cognitive rationale, administered on personal computers, mobile devices or gaming consoles. Interventions can be delivered individually or in group settings, with or without therapist supervision. Studies combining CCT with other non-pharmacological interventions (eg, physical exercise, brain stimulation) or with pharmacological interventions will be eligible as long as the CCT condition is the only key difference between the two groups (ie, study design allows to delineate the effect of CCT from the composite intervention). Studies will not be included if: (1) more than 50% of total intervention time was not CCT, (2) the intervention does not involve interaction with a computer (eg, passive viewing or recording of responses by an experimenter), (3) the CCT intervention is based on lab-specific rather than off-the-shelf hardware, which makes it less likely to be relevant to clinical practice.

### Types of comparators

Eligible control conditions include wait-list, no-contact and active (eg, sham CCT, recreational activities) control groups. Combined interventions (eg, pharmacological, physical exercise) will be eligible if provided similarly to both groups. All eligible controls in multiarm studies will be included.

### Types of outcomes

Eligible outcomes are change in performance from baseline to postintervention in non-trained measures of cognition (global or domain-specific), assessed through standardised neuropsychological tests or close equivalents (eg, a computer-based version of a common neuropsychological test). Additional outcomes include quality of life (standardised psychological well-being and quality of life questionnaires), mood (eg, clinical depression

rating scales), subjective cognition and daily function (patient or informant-reported activities of daily living questionnaires or standardised measures, for example, timed instrumental activities of daily living). Outcomes will be excluded if they were used as (or closely resemble) training tasks or exploratory in nature (ie, do not resemble common neuropsychological tests). In studies reporting more than one outcome measure per category, all eligible outcome measures will be included. The primary outcome will be overall cognitive performance.[5 10–12] Secondary outcomes are domain-specific cognitive performance, classified according to the Cattell-Horn-Carroll and Miyake framework,[13] subjective cognition, psychosocial functioning and daily function. Outcomes from longitudinal follow-ups will be included when available and meta-analytically investigated if appropriate.

### Search strategy

We will search MEDLINE, EMBASE and PsycINFO through the OVID interface for eligible articles. As this is an update of our previous systematic review,[5] the search will be limited to entries from 1 January 2015 and records from the updated search will be combined with eligible trials identified through the systematic literature search in the original version of the review. No restrictions on language or type of publication will be applied. The electronic search will be complemented by hand-searching the references of included articles and previous reviews as well as clinical trial registries. The full search strategy is shown in table 1. A systematic literature search was conducted on 14 May 2020.

### Study selection

Literature search results will be uploaded to a single Covidence library. Duplicates will be removed and articles identified from other sources will be added. Initial screening for eligibility based on titles and abstracts will be conducted by two independent reviewers. Full-text screening of potentially relevant articles will be conducted by two independent reviewers and disagreements resolved by consensus or by involvement of a third reviewer.

### Data extraction

Data will be extracted to a piloted Excel spreadsheet by one reviewer and a senior reviewer will check the data. Any disagreements will be resolved by consensus or by involvement of a third reviewer if necessary. If any additional information is needed, we will contact the corresponding authors of the studies. The following data items will be extracted:

► Study information: first author, year of publication, study location.
► Population: mean age, per cent male, mean Mini-Mental State Examination score (or equivalent), mean Unified Parkinson's Disease Rating score (or equivalent), disease stage (Hoehn & Yahr Scale or equivalent), disease duration, medication use, cognitive status (normal, subjective cognitive complaints or MCI).
► Intervention: type of CCT, programme used, training content, delivery format (supervised or unsupervised), total training duration (hours), session frequency (sessions/week), session length (minutes), total number of sessions, intervention duration (weeks), adjacent treatments.
► Comparator: type of control, control group activity.
► Outcome: name of test, summary data for each group (eg, mean, SD, sample size) at baseline and postintervention, cognitive domain.

Intention-to-treat data will be preferred if reported. Data will be extracted as means and SD for each time point if reported. If such information is not available, data in other formats (eg, mean change and SD) will be used if the article provides sufficient information to reliably calculate standardised mean difference. If these data are unavailable, authors will be contacted to obtain missing data.

### Risk of bias assessment

Risk of bias in individual RCTs will be assessed using the revised Cochrane Risk of Bias tool (RoB 2).[14] Low, high or some concerns risk of bias will be determined for each of the following domains:

| Table 1 | Search strategy |
| --- | --- |
| # 1 | ((cognit* or attention or neurocognit* or neuropsycholog* or memory or mental or reasoning or executive) adj2 (interven* or training* or rehabilitat* or remediat* or stimulat* or activit* or enhanc* or exercis* or retrain*)).mp. |
| # 2 | ((brain) adj2 (training* or rehabilitat* or remediat* or retrain*)).mp. |
| # 3 | (speed adj3 training).mp |
| # 4 | (video gam* or videogam* or wii or computer gam* or virtual reality).mp. |
| # 5 | 1 or 2 or 3 or 4 |
| # 6 | parkinson$.mp |
| # 7 | exp Parkinson's disease/ |
| # 8 | exp Parkinsonism/ |
| # 9 | 6 or 7 or 8 |
| # 10 | 5 and 9 |
| # 11 | limit 10 to yr="2015 -Current" |

1. Bias arising from the randomisation process.
2. Bias due to deviations from intended interventions.
3. Bias due to missing outcome data.
4. Bias in measurement of the outcome.
5. Bias in selection of the reported result.
6. Overall bias.

Studies with 'some concerns' or 'high' risk of bias in domains 3 or 4 will be considered as having some concerns or high risk of bias, respectively. Two independent reviewers will assess the risk of bias and disagreements will be resolved by consensus or consulting a third reviewer if necessary.

## Data synthesis

Analyses will be conducted using the packages metafor, metaSEM, robumeta and clubSandwich for R. Between-group differences in change from baseline to postintervention will be converted to standardised mean differences and calculated as Hedges' g with 95% CI for each eligible outcome measure. Pooling of outcomes across studies will be conducted using random-effects models, accounting for dependency structure of effect sizes within studies.[15 16] Sensitivity analyses for the primary outcome will be conducted by comparing results from multilevel and robust variance estimation models. Analyses of secondary outcomes will be contingent on the availability of at least three studies for analysis.

Heterogeneity across studies will be quantified using $\tau^2$ and expressed as a proportion of overall observed variance using the $I^2$ statistic.[17 18] Prediction intervals will be calculated to assess the dispersion of effects across settings.[19] Provided sufficient statistical power for investigations of heterogeneity,[20] potential moderators will be investigated using mixed-effects meta-regression models. The following moderators will be tested, if warranted: training content and type; control content and type; population (clinical or cognitive status); delivery format; training dose and frequency. Meta-regressions will not be conducted if heterogeneity in the overall model is negligible (ie, $\tau^2 < 0.01$) or when there are less than three studies within a planned subgroup. If warranted, potential interactions across moderators will be tested on an exploratory basis using multivariate meta-regression or network meta-analysis.

## Meta-bias(es)

Small-study effect will be assessed by visually inspecting funnel plots of effect size versus SE.[21] If at least 10 studies are available, small study effect will be formally tested using a multivariate analogue of the Egger's test,[22] that is, a meta-regression using SE as covariate. Subgroup analysis of the primary outcome will be conducted based on overall RoB 2 scores.

## Confidence in cumulative evidence

The strength of the evidence will be assessed and summarised qualitatively based on risk of bias for individual studies, precision of the effect estimates, heterogeneity across studies (including prediction intervals) and evidence for small study effects, with additional sensitivity analyses conducted if warranted.

## Patient and public involvement

Patients and/or the public will not be involved in this study.

## Ethics and dissemination

No formal ethical assessment or informed consent is required for this study. The findings of the study will be summarised in a manuscript that will be submitted for publication in a peer-reviewed scientific journal.

**Author affiliations**
[1]Academic Unit for Psychiatry of Old Age, Department of Psychiatry, The University of Melbourne, Melbourne, Victoria, Australia
[2]Department of Psychology, Umea University, Umea, Sweden
[3]Healthy Brain Ageing Program, Brain and Mind Centre, The University of Sydney, Camperdown, New South Wales, Australia
[4]Central Clinical School, Faculty of Medicine and Health, Charles Perkins Centre, The University of Sydney, Camperdown, New South Wales, Australia
[5]Department of Social and Psychological Studies, Karlstad University, Karlstad, Sweden
[6]Department of Neurology, Charité - Universitätsmedizin Berlin, Berlin, Germany
[7]Berlin School of Mind and Brain, Humboldt-Universität zu Berlin, Berlin, Germany

**Contributors** Guarantor: AL. Design and conceptualisation: HMG and AL. Data collection: HMG, MD and IL. Risk of bias assessment: HMG, MD and IL. Data analysis and interpretation: HMG, MD, IL, ASN, CF and AL. Drafting and revising the manuscript: HMG, MD, IL, ASN, CF and AL.

**Funding** This work was supported by a CR Roper Fellowship from the University of Melbourne provided to AL (2020-1), and by the Swedish Research Council (2017-02371) as well as the Swedish Research Council for Health, Working-Life and Welfare (2014-01654) awarded to ASN.

**Competing interests** None declared.

**Patient consent for publication** Not required.

**Provenance and peer review** Not commissioned; externally peer reviewed.

**ORCID iDs**
Hanna Malmberg Gavelin http://orcid.org/0000-0003-3256-9018
Magdalena Domellöf http://orcid.org/0000-0002-2348-1164
Anna Stigsdotter Neely http://orcid.org/0000-0003-3450-8067
Carsten Finke http://orcid.org/0000-0002-7665-1171
Amit Lampit http://orcid.org/0000-0001-6522-8397

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
