## [Reviewer comments · BMJ Open]

ARTICLE DETAILS

TITLE (PROVISIONAL)	Computerized Cognitive Training in Parkinson's Disease: A Protocol for a Systematic Review and Updated Meta-Analysis
AUTHORS	Malmberg Gavelin, Hanna; Domellöf, Magdalena; Leung, Isabella; Stigsdotter Neely, Anna; Finke, Carsten; Lampit, Amit

VERSION 1 – REVIEW

REVIEWER	Chris Vriend Amsterdam UMC, Vrij Universiteit Amsterdam, the Netherlands
REVIEW RETURNED	22-Jun-2020

GENERAL COMMENTS	In the proposed systematic review and meta-analysis protocol the authors aim to update the meta-analysis on computerized cognitive training (CCT) in Parkinson's disease (PD), that they published earlier.[1] They aim to extend on previous meta-analyses by not only studying the effects of CCT on global cognitive function (primary outcome), separate cognitive domains, and psychosocial and daily function, but also assess moderator effects of CCT efficacy, strength and quality of the evidence and ultimately pose recommendations for this field. My main question is however how this proposed meta-analyses relates to the systematic review recently published in Archives of clinical neuropsychology by Nousia and colleagues, that similarly investigated the computer-based cognitive training paradigms in neuropsychological performance in Parkinson's disease and also looked at risk of bias, etc. The authors should at the very least mention this study and how they compare to avoid redundancy. General - The authors are encouraged to mention the PROSPERO registration of the proposed analysis in the main text. Eligibility criteria - What is the rationale for excluding studies that exclusively assess CCT in PD dementia? Search strategy - What is the reason for excluding studies before January 1, 2015? If the authors aim to update the existing meta-analysis, would they not want to combine new and older literature? In their previous work the authors argued that the body of evidence from the seven randomized controlled trials was small, thereby limiting precision of the findings.[1] Do the authors expect to find enough trials to provide reliable results? Or might combining the evidence from their previous work with studies published after 2014 increase reliability of the findings?
---

	- The authors might consider to additionally search in 'grey literature' databases to enhance the reliability of the results and diminish potential publication bias. Data synthesis - The authors are encouraged to specify under what conditions moderators and interactions will be analyzed. 1. Leung IH, Walton CC, Hallock H, Lewis SJ, Valenzuela M, Lampit A: Cognitive training in Parkinson disease: A systematic review and meta-analysis. Neurology 2015, 85(21):1843-1851.
--	--

REVIEWER	Marco Cavallo eCampus University, Novedrate, Italy
REVIEW RETURNED	14-Jul-2020

GENERAL COMMENTS	The protocol titled "Computerized Cognitive Training in Parkinson's Disease: A Protocol for a Systematic Review and Updated Meta-Analysis" tackles an essential scientific and clinical topic, i.e. the investigation of efficacy and impact of CCT in PD through the analysis of its consequences on both cognitive and functional levels. Authors' protocol is pretty good, theoretically sound and methodologically complete. The planned study will represent an update of a previous study, as they declared to be interested in papers published from 2015 onwards. Just a general point should be clarified: Authors should better specify the main rationale of the protocol: if from previous reviews conducted in the field the efficacy of cognitive training in general had shown mixed and inconclusive results, why do they think that computerized-cognitive training specifically will be able to prove a more definitive efficacy (or a lack thereof)? It is reasonable to assume that only a very limited number of RCTs will satisfy the inclusion criteria of the protocol as computerized-cognitive training had been used only recently with patients affected by PD, and thus the main finding of the planned systematic review will probably be again "mixed and inconclusive results". Do authors really feel that it is already time to analyse systematically the studies conducted in the field instead of planning more robust studies in the field itself? Not sure about a positive answer to this question.
---

REVIEWER	Sara Bernini IRCCS Mondino Foundation, Italy
REVIEW RETURNED	21-Jul-2020

GENERAL COMMENTS	This study protocol reports a planned protocol for a systematic review research and update meta-analysis. The authors are waiting for registration number by the PROSPERO editorial team. Study design and methodology are exhaustive and of high quality. I think only a few clarification are needed. 1) In "Types of studies" authors explain that studies that provide neuropsychological assessment at baseline and post-intervention will be include and in "Types of outcomes" authors declare that changes in performance from baseline and post-intervention will be consider. I agree that an immediate post-intervention assessment
--

	allows to detect the presence of CCT effect, but other follow-up visit would be require to investigate the persistence of the training-related improvement and the impact of the intervention on the evolution of cognitive decline. I ask the authors to consider the importance of follow-up visits among the eligibility criteria. 2) About “Types of intervention” specify if the sessions can be individual and/ or group and if supervised and/or unsupervised by the therapist.
--	--

VERSION 1 – AUTHOR RESPONSE

Reviewer: 1

Reviewer Name: Chris Vriend

Institution and Country: Amsterdam UMC, Vrij Universiteit Amsterdam, the Netherlands
 Competing interests: none declared

#1. My main question is however how this proposed meta-analyses relates to the systematic review recently published in Archives of clinical neuropsychology by Nousia and colleagues, that similarly investigated the computer-based cognitive training paradigms in neuropsychological performance in Parkinson’s disease and also looked at risk of bias, etc. The authors should at the very least mention this study and how they compare to avoid redundancy.

We recognize that the systematic review by Nousia and colleagues had a similar focus on CCT in PD as our planned work; however, our review will address some important evidence gaps in the literature that have not been previously investigated. Specifically, Nousia and colleagues did not include a meta-analytical investigation and thus did not synthesize results across outcome domains, nor did they explore sources of heterogeneity. By investigating study and intervention design factors that could moderate CCT effects, our review will therefore provide a novel contribution to guide future research and practice in the field. In line with the reviewer’s suggestion, we have clarified this in our manuscript.

On p. 5, row 1-3, we write:

Finally, a recent systematic review focusing specifically on CCT reported evidence for cognitive benefits based on seven RCTs; however, no meta-analysis was performed nor were potential effect modifiers explored.

#2. The authors are encouraged to mention the PROSPERO registration of the proposed analysis in the main text.

This has been added to the abstract (p. 2, line 24) and main text (p. 5, line 23-24). Please note that due to the pandemic, PROSPERO is currently focusing on COVID-19 registrations and our submission was therefore published automatically and has not yet been checked by the PROSPERO team.

#3. What is the rationale for excluding studies that exclusively assess CCT in PD dementia?

This decision is based on previous meta-analytical evidence suggesting that CCT is less efficacious for people with dementia [1]. While it is difficult to infer whether this holds true also for individuals with PD dementia based on the current body of literature (e.g., Orgeta and colleagues [2] only identified one CT trial in PD dementia), inclusion of PD dementia would likely introduce clinical heterogeneity

that would preclude meaningful interpretation of the summary effects. Therefore, we have decided to exclude these studies from the present systematic review.

1. Hill NT, Mowszowski L, Naismith SL, et al. Computerized Cognitive Training in Older Adults With Mild Cognitive Impairment or Dementia: A Systematic Review and Meta-Analysis. *Am J Psychiatry* 2017;174(4):329-40. doi: 10.1176/appi.ajp.2016.16030360.

2. Orgeta V, McDonald KR, Poliakoff E, et al. Cognitive training interventions for dementia and mild cognitive impairment in Parkinson's disease. *Cochrane Database Syst Rev* 2020;2:CD011961. doi: 10.1002/14651858.CD011961.pub2.

#4. What is the reason for excluding studies before January 1, 2015? If the authors aim to update the existing meta-analysis, would they not want to combine new and older literature? In their previous work the authors argued that the body of evidence from the seven randomized controlled trials was small, thereby limiting precision of the findings.[1] Do the authors expect to find enough trials to provide reliable results? Or might combining the evidence from their previous work with studies published after 2014 increase reliability of the findings?

We thank the reviewer for this valuable comment. To clarify, we aim to combine the results from the updated search with the eligible studies identified from the systematic literature search conducted in our previous review. We apologize if this was unclear and have clarified this in the manuscript.

On p. 8, row 13-16 we write:

As this is an update of our previous systematic review, the search will be limited to entries from 1 January 2015 and records from the updated search will be combined with eligible trials identified through the systematic literature search in the original version of the review.

#5. The authors might consider to additionally search in 'grey literature' databases to enhance the reliability of the results and diminish potential publication bias.

We manually screened all Google Scholar entries citing the systematic reviews by Leung (2015), Lawrence (2017) and Ortega (2020) as well as the WHO International Clinical Trials Registry for potential studies. In addition, as mentioned on p. 6, row 14-16, we will follow up on potentially eligible protocol papers, conference abstracts and theses within our search results for completed but unpublished studies.

#6. The authors are encouraged to specify under what conditions moderators and interactions will be analyzed.

Consistent with Hedges and Piggot (2004, ref #20), power for meta-regressions will be estimated based on available heterogeneity, number of studies and sample size in the relevant subgroups as well as effect size difference. We will perform analyses when there are at least three studies per subgroup unless heterogeneity is negligible (i.e., nearly all variance is attributable to random error). We do expect however that power will be relatively low and thus have not set a rule of thumb threshold for conducting such analyses. Instead, we will report post-hoc power estimations to flag potential type I error for a range of clinically meaningful between-subgroup effect size differences.

On p. 11, row 16-18 we added:

Meta-regressions will not be conducted if heterogeneity in the overall model is negligible (i.e., $\tau^2 < 0.01$) or when there are less than three studies within a planned subgroup.

Reviewer: 2

Reviewer Name: Marco Cavallo

Institution and Country: eCampus University, Novedrate, Italy.

Competing interests: None declared

#7. Authors' protocol is pretty good, theoretically sound and methodologically complete. The planned study will represent an update of a previous study, as they declared to be interested in papers published from 2015 onwards. Just a general point should be clarified:

Authors should better specify the main rationale of the protocol: if from previous reviews conducted in the field the efficacy of cognitive training in general had shown mixed and inconclusive results, why do they think that computerized-cognitive training specifically will be able to prove a more definitive efficacy (or a lack thereof)? It is reasonable to assume that only a very limited number of RCTs will satisfy the inclusion criteria of the protocol as computerized-cognitive training had been used only recently with patients affected by PD, and thus the main finding of the planned systematic review will probably be again "mixed and inconclusive results".

Do authors really feel that it is already time to analyse systematically the studies conducted in the field instead of planning more robust studies in the field itself? Not sure about a positive answer to this question.

We thank the reviewer for the constructive feedback. Firstly, we would like to clarify that this update will also include studies that were published before 2015, that is, we will combine the results from the updated search with eligible studies identified in our previous review [1]. We apologize if this was unclear and have now clarified this in the manuscript, as a response to Reviewer 1, Item #4.

As has been demonstrated in recent systematic reviews [1-3] there has been an increasing number of studies published in the field at this point. We are therefore confident that an updated systematic review and meta-analysis will be of value for future research and practice in the field. Specifically, a major gap in the current literature is that no previous systematic review has investigated sources of heterogeneity and addressed the issue of moderators of CCT efficacy across different intervention designs and settings. Using a similar approach as in our previous systematic review of CCT in healthy older adults [4], we therefore aim to investigate under which conditions CCT is more likely to work, and equally important, to detect what is least likely to work. Thus, by incorporating investigation of potential CCT effect modifiers, and given the increasing number of studies in the field allowing for such investigation, this review will provide important knowledge to inform future intervention and study design in the field.

1. Leung IH, Walton CC, Hallock H, et al. Cognitive training in Parkinson disease: A systematic review and meta-analysis. *Neurology* 2015;85(21):1843-51. doi: 10.1212/WNL.0000000000002145.
2. Orgeta V, McDonald KR, Poliakoff E, et al. Cognitive training interventions for dementia and mild cognitive impairment in Parkinson's disease. *Cochrane Database Syst Rev* 2020;2:CD011961. doi: 10.1002/14651858.CD011961.pub2.
3. Nousia A, Martzoukou M, Tsouris Z, et al. The beneficial effects of computer-based cognitive training in Parkinson's Disease: A systematic review. *Arch Clin Neuropsychol* 2020;35(4):434-47.
4. Lampit A, Hallock H, Valenzuela M. Computerized cognitive training in cognitively healthy older adults: a systematic review and meta-analysis of effect modifiers. *PLoS Med* 2014;11(11):e1001756. doi: 10.1371/journal.pmed.1001756.

Reviewer: 3

Reviewer Name: Sara Bernini

Institution and Country: IRCCS Mondino Foundation, Italy Competing interests: None

#8. In “Types of studies” authors explain that studies that provide neuropsychological assessment at baseline and post-intervention will be include and in “Types of outcomes” authors declare that changes in performance from baseline and post-intervention will be consider. I agree that an immediate post-intervention assessment allows to detect the presence of CCT effect, but other follow-up visit would be require to investigate the persistence of the training-related improvement and the impact of the intervention on the evolution of cognitive decline. I ask the authors to consider the importance of follow-up visits among the eligibility criteria.

We fully agree that intervention effects in the medium- and long-term are important. These outcomes will be included and reported in the systematic review and analysed if appropriate, i.e., if a sufficient number of comparable studies are identified. We have clarified this in the manuscript.

On p. 8, row 8-9, we write:

Outcomes from longitudinal follow-ups will be included when available and meta-analytically investigated if appropriate.

#9. About “Types of intervention” specify if the sessions can be individual and/ or group and if supervised and/or unsupervised by the therapist.

We have clarified in our eligibility criteria that interventions can be delivered either individually or in group settings and with or without therapist supervision (p. 7, row 4-5). This approach will allow for investigation of the extent to which intervention delivery and settings could modify CCT efficacy.

VERSION 2 – REVIEW

REVIEWER	Chris Vriend Amsterdam UMC, Vrij Universiteit Amsterdam, the Netherlands
REVIEW RETURNED	13-Aug-2020

GENERAL COMMENTS	The authors have satisfactorily answered my queries. I have none further.
---

REVIEWER	Marco Cavallo eCampus University, Novedrate (Italy)
REVIEW RETURNED	13-Aug-2020

GENERAL COMMENTS	The protocol titled “Computerized Cognitive Training in Parkinson’s Disease: A Protocol for a Systematic Review and Updated Meta-Analysis” tackles an essential scientific and clinical topic, i.e. the investigation of efficacy and impact of CCT in PD through the analysis of its consequences on both cognitive and functional levels. The planned study will represent an update of a previous study, as the authors declared to be interested in papers published from 2015 onwards. Authors’ protocol is appropriate, theoretically sound and methodologically complete.
---

REVIEWER	Sara Bernini
-----------------	--------------

	IRCCS Mondino Foundation, Pavia, Italy
REVIEW RETURNED	13-Sep-2020
GENERAL COMMENTS	I find the article satisfactorily revised.